# Peri-Urban Organic Agriculture and Short Food Supply Chains as Drivers for Strengthening City/Region Food Systems—Two Case Studies in Andalucía, Spain

**Carolina Yacamán Ochoa [1],*** , **Alberto Matarán Ruiz [2]**, **Rafael Mata Olmo [3]**,
**Álvaro Macías Figueroa [2] and Adolfo Torres Rodríguez [4]**

[1] Department of Geography, Faculty of Geography and History, Complutense University of Madrid, 28040 Madrid, Spain
[2] Department of Urban Planning and Spatial Planning, Faculty of Science, University of Granada, 18071 Granada, Spain; mataran@ugr.es (A.M.R.); alvaritoxere@correo.ugr.es (Á.M.F.)
[3] Department of Geography, Faculty of Philosophy and Letters, Autonomous University of Madrid, 28049 Madrid, Spain; rafael.mata@uam.es
[4] Department of Sociology, Faculty of Political Sciences and Sociology, University of Granada, 18071 Granada, Spain; atorresr@ugr.es
* Correspondence: cyacaman@ucm.es

**Abstract:** Discussions on food security in the Global North have raised questions about the capacity of peri-urban organic agriculture to provide sufficient healthy food for the urban market. Dealing with food security requires more attention to how to protect peri-urban organic farming systems from urban pressures while strengthening the sustainability of local food systems. Given that short food supply chains (SFSCs) have been proven to be effective at reconnecting people with food production, this study focuses on identifying the barriers that hinder their development and the opportunities derived from the comparative advantage provided by their urban proximity. This study is based on documentary and empirical research addressing food supply chain characteristics in the organic sector. This study is focused on Mediterranean peri-urban agriculture, where, historically, there have been close relationships between the city and the countryside. These relationships are based on the fact that many cities are traditionally located next to areas of high agricultural activity, where a wide variety of vegetables is produced almost continuously due to the relatively mild winter climate. This study deals with two medium-sized metropolitan areas in Andalucía in the south of the Iberian Peninsula—the coastal city of Málaga, which is of a tourist-residential nature, and the inland urban agglomeration of Granada. Our research shows, when compared with other studies, that the local organic food sector seems to have great potential to find innovative solutions based on a collective approach, local embeddedness, and collective knowledge and by prioritizing horizontal and sustainable processes at the local/regional scale.

**Keywords:** mediterranean farming systems; urban and metropolitan region; logistics; distribution; networking; food security; food chain stakeholders; local embeddedness; social innovation; food security

## 1. Introduction

Factors such as urban sprawl, intensive fossil fuel use, climate change and the globalized agri-food system increase the vulnerability of metropolitan areas with regard to food security. For example,

the massive rural exodus to the cities and the subsequent population concentration which intensified during the second half of the twentieth century drove a large number of people away from the production of foodstuffs and has made them dependent on increasingly distant regions to ensure their food supply [1,2]. At the same time, large Mediterranean peri-urban farms face an uncertain future because of a loss of profitability, little guarantee of generational handover on small-scale farms [3], the struggle for water and land, competition in the labor market [4], and an increase in artificial surface areas at the expense of the traditional coastal and inland landscapes [5]. Rising temperatures are also altering the capability of agricultural land to continue performing its role, which is intensified by the concentration of economic activities and the intensive occupation of fertile land by transport infrastructure [6,7]. More extreme situations, such as the COVID-19 (Corona Virus Disease 2019) pandemic and the restrictions on the movement of people and goods in a growing number of countries, are putting major strains on local, regional, and global supply chains, testing the resilience of food systems [8], revealing the vulnerabilities of short and regional supply chains, and jeopardizing access to certain fresh produce traditionally supplied in some regions (closure and restrictions on farmers' markets, travel constraints, problems in connecting the different operators in the supply chain, etc.). In short, in the context of local and global challenges of urban sprawl and limited resources, it is crucially important to guarantee access to safe food products, efficient organic production and agri-food logistics to resolve issues regarding food security [9], and to strengthen every aspect of the sustainability of foods.

If peri-urban agriculture is to be considered as a lever for restoring social and cultural ties between cities and the agricultural space which surrounds them, it needs special recognition to allow the reduction in converging pressures that jeopardize its future. In this context, as is already happening with organic agriculture, there is a resurgence of interest in peri-urban agriculture as a result of its ability to improve the welfare of citizens, including its contribution to food security [10–12] and its potential to ensure the regional embedding of the agri-food system [13]. As highlighted by Cerrada-Serra et al. [14], these issues are especially significant for Mediterranean cities and regions, which have strong historical ties to their agricultural surroundings. However, beyond recognizing the importance of organic and peri-urban agriculture, it is necessary to increase the amount of research on estimating and planning the agricultural system at different levels [15–19]. Similarly, more knowledge is needed to foster farmer cooperation on food logistics, given that this issue in the food chain has a direct effect on the economic viability of small-scale farms, the price of food, and consumer satisfaction [20].

In this framework, our research is focused on the hypothesis addressed in recent studies [21,22] and sustained by other authors [13,23–25] that short food supply chains (SFSCs) have the potential to improve the viability of peri-urban organic agriculture, promote sustainable farming systems and reduce the vulnerabilities of the local food system on multiple levels. SFSCs can do this through their role as catalysts of initiatives which activate endogenous resources, encouraging the creation of added value in foodstuffs. They also promote regional embeddedness, contributing to local economic development. Moreover, the SFSCs boost the creation of networks, reduce the impact of agriculture on the environment, and improve democratic decision making in regional food systems.

This paper is focused on identifying the barriers that limit the scaling up of the SFSCs, and the opportunities derived from the comparative advantage of peri-urban organic agriculture due to its proximity to urban markets and its capacity to reconcile different dimensions of sustainability. Combined research analysis of primary and secondary sources has been carried out to achieve this objective. The specific purpose of this study is to widen knowledge on small peri-urban producers involved in the SFSCs, identify their main needs regarding scalability, simplifying logistics, reducing their ecological footprint, and increasing visibility in the urban market. In short, this will lead to the discovery of the factors that allow small-scale farms to enjoy increased income. From a wider perspective, the global objective of this paper is to improve the understanding of supply chain organization at a city/region scale to offer solutions which help to reduce food vulnerability in metropolitan regions and restore the synergy between the countryside and cities.

This paper is organized in the following manner: after the introduction, Section 2 identifies the challenges to development faced by the SFSCs using a collaborative approach. Special attention is given to product development, market access, logistics and distribution, identification of success factors and obstacles. Section 3 describes the method used to carry out the research and introduces the territorial context of Andalucía, looking in depth at the urban planning processes and the fragmentation of the land, which have seriously modified the agricultural system of the Vega of Granada in the metropolitan area of Granada and the Mid-Guadalhorce Valley region in the agricultural urban greenbelt of Malaga. Section 4 presents key results obtained from interviews with small organic producers and retail establishments. Section 5 discusses the results of the contributions in Section 2 and the data from Section 4 to orientate appropriate developmental strategies for the SFSCs. Finally, this paper concludes with a critical reflection on key aspects to improve the scaling up of the SFSCs, among which is the need to encourage a change in values for the promotion of a regionalized agri-food system. This highlights the need for a cooperative approach to improve collective knowledge, improve the production of local identity-based foods, increase public–private or urban–rural partnerships, improve logistical bottlenecks and, finally, prioritize sustainable horizontal processes at a local/regional scale.

## 2. Literature Review on Drivers that Help Scale up Short Food Supply Chains

Taking the globalization and the liberalization of agri-food markets into consideration, peri-urban organic agriculture must adapt to make the most of the contemporary urban demand for healthy food. Recently, there has been a major change in public policies on the way in which food security is dealt with in recent decades by focusing on building regional food system resilience at all levels. From this perspective, research on SFSCs has shown that, economically, these channels can provide greater income for local producers, given that they can retain a larger amount of the profit, which would otherwise be absorbed by intermediaries. However, as stated by Malak-Rawlikowska [24], because of the small quantities demanded in the different selling points, they can end up making a larger environmental impact, because their efficiency is lower than that of the massive conventional commercialization systems, or be less profitable. This forces producers to look for support on social and technological innovation for the deployment of SFSCs [26]. Scientific literature is looking at these issues in depth, providing innovative solutions to help small-scale farmers develop methods that are more successful and sustainable in environmental, economic, and social terms. Therefore, for regional agri-food systems to reach their full potential, the producers must find a way to increase scalability (in terms of the number of farmers involved and the volumes sold) in order to sell their products to a greater number of consumers while introducing environmental sustainability criteria [27].

### 2.1. Challenges to Improving Regional Food Logistics and Distribution Using a Collective Approach

Logistics and distribution are two aspects which are essential for the success of SFSCs. Therefore, it is necessary to define the roles involved in the food supply chain before analyzing its associated problems. Mentzer (2001) [28] highlights the definition provided by the Council of Logistics Management (1998) "Logistics is that part of the supply chain process that plans, implements, and controls the efficient flow and storage of goods, services, and related information from the point of origin to the point of consumption in order to meet customers 'requirements". According to Mittal, Krejci and Craven (2018) [27], efficient logistics management must consider three key aspects: (i) transportation (movement of inventory from point to point in a supply chain); (ii) warehousing (activities involving the physical locations where inventory is stored, retrieved, assembled, and packed for distribution); and (iii) inventory management (monitoring and deciding how much inventory to store, what is in stock, and how inventory should be stored).

When using a cooperative approach to make the logistics of regional food supply efficient, the following key areas need to be considered: (i) improve the efficiency of the whole supply chain through the shared use of vehicles and by creating shorter routes between the production zones and the local consumption points; (ii) strengthen the relationships in the supply chain, mainly with

institutions to encourage public purchasing and small-scale commerce; (iii) optimize logistics to cope with inefficiency on a regional level through the aggregation of multiple small-scale farms to improve efficiency and reduce costs; (iv) carry out, wherever possible, joint planning between several small-scale farmers for a range of diversified products, ensuring year-round supply; (v) secure finance to invest in new technology (e-commerce and GIS tools) [21,29].

### 2.1.1. Managerial Capability and Supply Chain Infrastructure for Aggregating Products for Regional Markets

In Spain, small producers do not have a major role in direct supply through certain distribution channels, such as small neighborhood shops and collective catering (school canteens, universities, hospitals, care homes, etc.). This is due to not having an economy of scale and the higher production costs per unit for the SFSCs [24]. Therefore, to improve the traceability and distribution capabilities of small producers, particularly organic ones, in the regional food system, social innovation strategies are used. As stated by Beckeman et al. [30], they focus on carrying out grouped planning of the production of small-scale farmers in order to meet the demand of larger customers, for example, through food clusters and food hubs.

Food hubs—a type of organizational model—act as intermediaries between the local consumption points and the small producers in areas such as marketing, distribution and sales, with the objective of aggregating products from small-scale local farmers to enhance their economic viability [22]. Based on this definition, the efficiency of the market relationships of the small producers can be improved, by providing middleman services, which ultimately allows small producers to achieve a business model that is more solid and better orientated to meet the growing demand of local produce. By aggregating products from local small producers they can reduce transaction costs and increase their negotiating power [21]. According to Perrett et al. [31], this type of collective infrastructure provides better guarantees on hygiene measures, ensuring consumer safety, which is vital to maintain their role in food distribution, as well as allowing cost savings by resource sharing. Furthermore, they can have a positive impact on the viability and the stability of small-scale agriculture through the following ways: (a) investment capability; (b) joint planning to satisfy the increase in supply of the volume and produce range; (c) support in tasks like efficient marketing and product development, customer service, commercial negotiation, economic management, etc.; (d) reduction in expenses associated with logistics and distribution (frequency of deliveries, dispersion of sales points, cold chains, storage facilities, processing and distribution, etc.) [22]. Nevertheless, the transformative potential to support the emergence of food democracy is conditioned by the internal governance structure which is required for their business management and by the social responsibility agreed on by its members. Accordingly, the farmers must play a relevant role, not only by producing healthy food, but especially in enhancing the role of small producers in decision making, shared risk and profit sharing. Therefore, by focusing on a network approach, cooperative food hubs which prioritize small producers and make a commitment to the local community present a paradigm shift in regional food distribution. As a result, they can make the most of the experience and know-how of their members, promoting and sharing commercial skills and grouping together production to better respond to local demand.

### 2.1.2. Efficient Transport

The main challenge presented by the distribution of local food in regional agri-food systems is organizing the routes from the production zones to the local consumption points as efficiently as possible. This organization takes into account the supply demands and the reduction in time and fuel costs, as well as the ecological footprint of the whole process. However, Bence [32] argues that the lack of technological, financial and organizational innovation held by the small organic producers, the dispersion of sales points and limited storage capabilities are factors which end up eliminating the environmental benefits and the profits which are provided by the geographical proximity of the SFSCs. Nevertheless, these challenges can be overcome by focusing on cluster building,

logistic network integration of local supply chains and using internet-based solutions. As stated by Bosona, and Gebresenbet [33], by using internet-based solutions businesses can guarantee better control of logistic management by building up detailed information on location analysis (mapping and clustering producers and determining the optimum location of collection centers) and route analysis (creating the best routes for product collection and distribution, simulating route distance and delivery time), reducing transport distance and time. Accordingly, digital technology can be an ally in improving the competitivity and traceability of the small producers by ensuring better flexibility, productivity, cost efficiency and better levels of control for logistics activity [34].

### 2.1.3. Exchange of Knowledge, Skills, and Information

Not all SFSCs have the same impact in the agricultural sector or upon aspects related with improving their scaling up. As emphasized by Yacamán et al. [21], the collective approaches are the ones that generally show a greater capability for social, environmental and economic transformation. For example, food clusters and food hubs are designed as spaces that enable small famers to collaborate and generate new ideas with bottom–up training, as well as being laboratories for product development and fostering creativity to address market demand [22]. In turn, the organizational and physical proximity of different agents in the food chain reduces costs and increases the regularity of the exchanges collective skills, and facilitates the creation of trust-based relationships [35]. These are aspects which are essential for the creation of cooperation networks between producers, and for the establishment of community ties—both of which are necessary for promoting social innovation in the agro-ecological sector [36] and for building a cooperative system which is able to secure the production and provisioning of food. In short, the resilience of the agri-food system is strengthened by improving collective knowledge, which is also key to the empowerment of small-scale farms. For Swagemakers [37], local producers can improve their social, economic and environmental abilities to face changes and pressures through the creation of localized self-governance structures and the promotion of agricultural good practices.

### 2.1.4. Market Differentiation and Consumer-Driven Changes

The growing interest in the consumption of traditional, local, and organic foods is contributing to the exponential increase in SFSCs. Going against the progressive introduction of "food from nowhere" [38] and the homogenization and standardization of production practices, there is a reaffirmation of the need to improve traceability and transparency regarding production methods to evaluate the quality and regional specificities of foods. In this context, the need to improve production differentiation strategies and increase regional anchoring of foods has gained momentum on a European level. To this end, as stated by Sanz and Muchnik [39], a network of committed local agents is required to cooperate mutually in a specific regional area to achieve this goal. Examples such as the European certifications, the Protected Designations of Origin (PDO) and the Protected Geographical Indications (PGI), local labels from the agricultural parks, participatory guarantee systems, and eco-labels have improved the identification of identity-based foods and have contributed to the strengthening of good agricultural practices.

Other similar experiences, such as the different participatory patrimonialization processes of peri-urban agricultural landscapes in Spain and France, aim to strengthen ties between food, identity and local communities. For Mata and Yacamán [40], this approach allows the history of the products, the intangible knowledge of traditional cultivation techniques and the agricultural landscapes to be discovered and made visible as a "living heritage". For these authors, all these anchoring strategies allow the attributes of the Mediterranean agricultural systems to be connected to the agricultural heritage, highlighting the qualities of local foods, organoleptic specificity, freshness and seasonability [40]. There are further positive aspects to these strategies, such as the fact that they are not industrial and have a greater commitment to the environment and culture, and that they positively connect food with the territory. As a result, the agricultural landscapes stop being passive and start to play a strategic role

in increasing the added value of the foods through the historical accumulation of local resources and actors. In addition, territorial governance is necessary to enhance the market value of autochthonous varieties by enhancing differentiation related strategies that value and positively condition the added value of products and the local economy [4]. In short, as Boons et al. [41] argued, four recommendations for making a successful market introduction and differentiation are needed: the implementation of socio-technical experiments, establishment of a broad network of actors, the building-up of a shared project vision and the creation of reflexive learning processes.

In general terms, and according to the bibliography which has been analyzed, we can state that in order to improve the scalability and resilience of SFSCs, it is necessary to improve the distribution and commercialization processes of locally cultivated products through the strengthening of collective infrastructure, and social and technological innovation. Among the drivers indicated in the literature for strengthening the resilience of regional agri-food systems, certain studies stand out the need to strengthen cooperation among the small producers involved at a regional scale. There are other key elements such as improving the collective knowledge of small producers, strengthening trust among them, and increasing territorial anchoring strategies to improve their ability to generate added value for the local foods and to differentiate themselves from the large-scale organic sector.

## 3. Materials and Methods

*3.1. Introduction of the Two Case Studies. The Metropolitan Regions of Granada and Malaga: Two Threatened Mediterranean Landscapes with Great Patrimonial and Agricultural Value*

### 3.1.1. The Metropolitan Region of Granada

The Vega of Granada is an extremely representative patrimonial, historical, and cultural symbol of the Mediterranean agricultural landscapes. This bioregion forms an ecological corridor linked to the river Genil and its different tributaries. It has an extensive surface area of 87,230 hectares [42,43], including 16,000 hectares of irrigated crops [44]. All of the previously mention features provide important ecosystem services. The historic irrigation system of the Vega of Granada, like in other parts of the agricultural system of the metropolitan region, is based on a structure of irrigation channels which are over 1000 years old and give life to lands which boast great fertility in a regional context marked by the advance of the semi-arid landscapes associated with less fertile soils than those of the plains [45].

Nevertheless, for decades, the deterioration and the accelerated disappearance of a large part of the traditional agricultural landscape of the Vega of Granada has taken place, as in many other metropolitan regions. This is a consequence of the general agricultural crisis [46], which includes, among other things, the intensification of industrial crops and the abandonment of traditional systems. This is also the result of the processes arising from the organization of its city and the metropolitan area that requires high levels of mobility and urban land, meaning a radical transformation in both the operation and spatial organization of the Vega. The location of the Vega within the metropolitan area of Granada (575,889 inhabitants, 2018) is facing pressures from excessive urban land occupation and fragmentation. Feria [47] analyzed different metropolitan regions of Spain and indicates that even though Granada has one of the first Metropolitan Regional Plans, it has one of the highest levels of real estate growth, together with Murcia and Las Palmas. This urban sprawl phenomenon can be explained by a weak participatory planning framework that did not consider the overall conservation of fertile land.

Despite the problems resulting from the advance of urban sprawl and the incorporation of agro-industrial crops, Matarán and Yacamán [48] argued that, the Vega of Granada has preserved a major part of its fertile land and its traditional productive identity, characterized by polyculture (vegetables, fruits, silk, flax, and hemp, among others) and small-scale farmers (more than 90% of the plots measure less than one hectare). Considering the renewed approaches to agri-food planning, the role of alternative food networks in Granada is notable compared to other cities in Andalucía.

Thanks to this, in recent decades, there has been an increase in initiatives on SFSCs at a bioregional scale (consumer groups, farmers' markets, organic grocer's, etc.), which seek to reconnect the fertile peri-urban land with the city. This means a new opportunity for preserving small-scale periurban farms and to maintain extensive irrigated land near the city.

### 3.1.2. The Metropolitan Region of Málaga

The agricultural area around Málaga and its urban region are similar, but with some distinctive features. The most significant changes produced in the area of influence of the city and the tourist/residential area known as the Costa del Sol have meant, as in many other areas on the Mediterranean coast, an intense tertiarization and economic dependence on tourism. As highlighted by Galacho [49], at the same time as losing much fertile land close to the traditional villages for tourist development, new areas of intensive industrial agricultural production have been developed to meet global agri-food demands. Likewise, the city center and some tourist nucleuses have been very attractive for job seekers responding to the demand for labor, taking precedence over the surrounding rural areas. The labor required by these areas has generally been for low-skilled workers, for construction and the service industry. This has radically transformed the socio-occupational profile of a large area, which goes beyond that which is habitually considered as being peri-urban, establishing a new framework of country–city relationships in the urban tourist region [50]. All of this, added to deficient metropolitan planning, which has not been concerned with protecting fertile land [47], explains why, since 1999, 55.86% of the agricultural land has been destroyed, going from 89,684 agricultural hectares in that year to 39,589 hectares in 2009, an average annual loss of 5.6% of the agricultural surface area (agricultural Spanish census 2009). This dynamic is similar, though more intensive, to that of Granada.

In this region, one of the most important agricultural areas is the Mid-Guadalhorce Valley region, which is now considered as the pantry of Málaga. The Mid-Guadalhorce Valley— traditionally known as Hoya de Malaga—is located at least 30 km from the city (574,626 inhabitants at the beginning of 2019), occupying a strategic location for the food supply of the metropolitan area (1,282,021 inhabitants). The research carried out by Comino et al. (2014) [51] on the changes in land use in an area of the Guadalhorce Valley show how the peri-urban planning processes from the central metropolis (urban occupation, logistics and industrial use, and the fragmentation of plots by transport links), along with coastal overcrowding, have meant that the valley is functioning as an area of urban expansion through land deals and land rezoning. The same study shows that the peri-urban planning process has taken place in parallel with the economic transformation of agricultural activity in recent decades, characterized by the move from traditional rainfed crops and extensive pasture land to an irrigated agricultural area specialized in citrus fruits [51]. Despite this, a unique agricultural landscape is partly maintained, with valuable constructed heritage in the form of small estates and laborers' cottages, and a large area which is still cultivated with a wide diversity of crops. There are vegetables, fruit trees and especially citrus fruits in the river plains of the river Guadalhorce, rainfed crops (olive groves and herbaceous plants) on the hillslopes and the recent introduction of tropical crops, which are highly profitable.

In spite of the tensions which have been described, the Mid-Guadalhorce Valley manages to maintain its agricultural character, thanks to, among other things, the development of innovative activities based on the SFSCs which are coordinated around *Guadalhorce Ecológico*. This cooperative is made up of a group of organic producers from small-scale farming backgrounds who cultivate fruit and vegetables in villages in this region, and have formed a network of markets for organic produce, a home delivery system and a direct commercialization structure for shops and restaurants which includes a wide variety of year-round seasonal produce.

### 3.2. Materials and Methods

This study is part of a series of studies of research carried out in Spanish metropolitan areas in the framework of a Spanish operational group called "Agri-food Hub: Feeding people, managing

territories". In this research project, the peri-urban organic agriculture sector was selected as an area of interest for three main reasons [21]: firstly, because peri-urban agriculture in general, and the organic section in particular, constitute important sectors in the proximity of urban agglomerations, supplying multiple goods and services that are demanded by the urban society. This field of research and action therefore endows peri-urban organic agriculture with a leading role in rearranging the relationships between country and city; secondly, because geographical proximity to urban areas constitutes yet another opportunity to strengthen an emerging urban food culture that prioritizes consumption of "zero-miles" organic produce provided by small-scale farms operating close to the cities; and finally, because it has been seen that it is very difficult to design strong and sustainable agri-food strategies aimed at relocating the food system if the local production and the corresponding supply chain is not orientated towards the nearest urban market.

The methodological approach is based on the distributive method and uses the production and analysis of survey data as a technique. The survey technique has been chosen as it allows quantitative data to be systematically obtained with a questionnaire, as well as permitting the statistical treatment of this data, which favors compared analysis with other similar studies [21,22]. This study was carried out in two specific metropolitan areas (Granada and Málaga) and two subsamples are identified in each area: one, producers in peri-urban organic agriculture; and two, small retail traders as points of sale.

This study was carried out in the metropolitan areas of Granada and Málaga, in the autonomous region of Andalucía (Spain). Both are regions with a notable presence in the small-scale organic horticulture sector (average of between 2 and 4 hectares) and have been in existence for a longer period, with more experience and greater projection related to food supply in local agri-food systems when compared to other similar cities in Spain. Furthermore, this choice of localities reflects the manner in which peri-urban agriculture responds to the demand for food in areas subjected to urban sprawl.

Sampling and Data Production

The research has been developed in two stages [52]. Firstly, the analysis of secondary sources has led to a bibliographical and documentary review to identify the regions selected and analyze their features, as presented in the previous section. The second stage is on applied research, using a survey technique to obtain primary data for the fixed objectives of the research on the widening of knowledge on the small peri-urban producers involved in the SFSCs. The data from this second stage was gathered through interviews and observations of the research team members during the field visits between November 2018 and January 2020. It is an exploratory study that considers the organic producers and the retail traders as the population universe.

The sample for the first group was established using two criteria for the main selections: small- to medium-sized farms (between 2 and 4 hectares) whose produce is distributed mainly through a short chain. In the absence of a census on these types of producers as a sample base, the process followed for the selection of the final sample units (farmers) was the snowball technique. This achieves the completion of the typology of peri-urban organic producers in both areas as strategic representation criteria rather than statistical representation. For the creation of said typology, statistical information on organic production was provided by the Andalusian Department of Agriculture, Fisheries and Rural Development.

To conduct the sampling of the farms, we considered previous studies and the knowledge of the University of Granada research group [44] and Landscape and Territory in Spain, Mediterranean Europe and Latin America research group of Autonomous University [53] complemented by the "snowball technique", asking the first producers chosen to identify other producers in the selected peri-urban areas in order to consider almost 100% of the small organic producers. In total, 42 interviews with questionnaires were carried out in Malaga and 15 in Granada, giving a total sample number of 57 producers. It is important to highlight that the sample cannot be considered as being representative of the combination of the population of peri-urban organic producers in Granada and Malaga. For the statistical representation of a sample such as this, a complete up-to-date database would be required

of the small-scale farms which mainly trade with SFSCs, and this type of database does not exist. Nevertheless, in agreement with the opinions of the farmers and the knowledge of the research group, we consider that the sample available is sufficient to allows us to improve our knowledge of this sector and to present valid conclusions on the regional agricultural sector as there are few farms which meet these criteria (priority participation in SFSCs and small-scale in peri-urban areas).

In the sample (Table 1), small- and medium-sized peri-urban farms which focus their production on horticultural crops and fruits are represented. This choice was made, firstly, because, historically, they have supplied the urban market, and secondly, because they have been forced towards intensive crops with a higher added value. This has been caused by the urban pressure (high land prices, urban planning, fragmentation of agricultural area, etc.) which they are subjected to. The majority of the products traded are produced by the farmers themselves (close to 90% of those interviewed.).

**Table 1.** General characteristics of the sample for organic producers.

| Metropolitan Area | Málaga | Granada |
|---|---|---|
| Number of farms | 42 | 15 |
| **Main type of production (%)** | | |
| Vegetables | 23.8 | 60 |
| Fruits | 71.4 | 13.3 |
| Others (meat, milk, olive oil, wine) | 4.8 | 26.7 |

Source: created by the authors.

The inclusion of small shops in this study is justified because they are an important channel for the promotion of direct sales of local produce. The data obtained helps us discover which aspects must be improved on by the small producers to strengthen the supply of this channel. The following types of establishments have been taken into account when selecting the sample: (i) municipal market stalls; (ii) city center shops; (iii) shops in lower–middle–income neighborhoods; (iv) shops in high–income neighborhoods; (v) shops specialized in organic produce. In total, 47 interviews were carried out in Malaga and 16 in Granada, in each establishment, for a total sample of 63 establishments (Table 2).

**Table 2.** Number of interviews for each sample type in each study area.

| Area/Sample | Producers | Retailers |
|---|---|---|
| Granada | 15 | 16 |
| Málaga | 42 | 47 |
| Total | 57 | 63 |

Source: created by the authors.

Following the objectives of the survey and previous research [21,22], we prepared a questionnaire for small organic farmers according to the typology of small- to medium-sized farms (employing from two to five people full-time throughout the year). The interview questionnaire had three sections that included key aspects such as production, commercialization and produce value. The questionnaire given to the retailers involved key aspects of who supplies them with vegetables and whether they purchase locally.

The univariate data analysis carried out is exploratory and descriptive (a detailed description of the distribution of the answers of each of the variables, generally using percentages) and aims to study the frequency distribution of the variables contained in the questionnaires. This provides information on the specific values which the selected variables have provided to increase our knowledge of the small peri-urban producers involved in the SFSCs. This information is used to identify their main needs for the improvement of scalability, the simplification of logistics, the increase in added value and their visibility in the urban market.

## 4. Results

### 4.1. Characteristics of the Sample

The data was collected from a sample of 57 organic horticultural producers (Table 3; more than 50% of the small-scale farms are less than 4 hectares in size). The data on labor resources and employment is remarkably similar in both of the metropolitan areas surveyed. The labor resources expressed in annual work units (AWU) per business unit oscillate, on average, between 1.33 AWU in Malaga and 1.87 AWU in Granada. The low level of labor on the small-scale farms, despite being labor-intensive crops (fruit and vegetables), is explained by the fact that mainly small- and medium-sized farms are represented in the sample. In total, 90% of those interviewed in Málaga state that they have not received any state aid for setting up their farms, and the data is similar in Granada (80%), which shows how little state support exists for this strategic sector in Andalucía.

**Table 3.** General characteristics of the sample.

| Metropolitan Area | Málaga | Granada |
|---|---|---|
| Number of farms | 42 | 15 |
| **Employment** | | |
| Average AWU per farm | 1.33 | 1.87 |
| From 0 to 1 AWU per farm * (%) | 76.3 | 33.3 |
| From 2 to 5 AWU per farm * (%) | 14.3 | 46.7 |
| More than 5 AWU per farm * (%) | 9.5 | 20 |

*Annual work unit (AWU) equivalent to full-time employment. Source: created by the authors.

Sales are mainly to the end consumers (individuals or organizations), and, to a smaller extent, to intermediaries such as distributors and purchasing centers. All of the farmers interviewed trade through different types of SFSCs (farm sales, internet sales, box schemes, farmers' markets) and are concentrated as indicated by the sales percentages in Table 4. In Malaga, the main commerce channels are on-farm sales and farmers' markets, which are a clear commitment of this region. In Granada, box schemes predominate in the first place, secondly, direct sales to restaurants and small markets, and third, e-sales in farmers' markets, which have also been followed in the example of Malaga. In Granada, on-farm sales are more commonly used in extensive farming mainly export oriented [44]. On the contrary, direct sales to public food procurement (schools, universities, councils, hospitals, etc.) could prove to be an opportunity for improving small-scale farm incomes and the sustainability of the regional agri-food system. However, the sales figures are anecdotal in Malaga (4.8%), while in Granada the percentage is much higher (53.3%). In Malaga, as in many other areas, this data is the result of the supply of services being concentrated in the hands of just a few suppliers [54] and supply protocols which are difficult for small-scale farmers to follow. Nevertheless, in Granada, there is an organic canteen in a public school that buys local produce and at least six initiatives for organic canteens for children self-managed outside the schools themselves, which buy produce directly from local producers. Despite the fact that more than half of the producers interviewed belong to some type of association, network or cooperative, the regional infrastructure for joint distribution of organic produce is insufficient to meet demand.

**Table 4.** Structure of sales by distribution channel.

| Supply Chains | Málaga | Granada |
|---|---|---|
| **Type of short chains used (%)** | | |
| On-farm sales | 39 | 13.3 |
| Box schemes | 7.3 | 53.3 |
| Internet sales | 0 | 6.7 |
| Direct sales: restaurants and shops | 7.3 | 33.3 |
| Farmers' markets | 31.7 | 40 |
| Direct sales: Public sector catering | 4.8 | 53.3 |
| **Types of long chains used (%)** | | |
| Sales to intermediaries | 24.4 | 6.7 |
| Wholesale market (*Mercas*) | 26.8 | 20 |

Source: created by the authors.

### 4.2. Farmers' Opinions on SFCSs

In order to study the perception that small peri-urban producers have about SFCS in terms of their ability to improve the economic viability of their farms, we analyzed those channels that they consider of greatest interest to increase the volume of sales of food products and also the income obtained by reducing intermediaries. According to the data obtained from the subsample (Table 5) for Málaga, the SFSC which has been evaluated as most interesting is farmers' markets (30%), which, as we have mentioned, is a hallmark of this geographical area. In Granada, the most highly valued direct channel is box schemes. The lack of interest in increasing direct sales to restaurants and small shops is surprising, despite being territories with an important tourist weight, where they are an essential economic sector. This lack of interest can also be seen in the supply of public food procurement. It is tied to the lack of ability to supply the demand of the channels which need high levels of daily distribution of widely diversified produce. Sales through web pages or e-commerce continue to be a little-known strategy for this sector despite the growing potential of this channel. The potential of this channel has increased greatly due to the emergency brought about by the COVID-19 pandemic. However, if their perception of increasing their sales with the wholesale and the distribution sector is analyzed, both regions coincide in the low valuation due to their low capacity to negotiate prices, although they show a clear preference in Malaga for selling tropical crops to distribution companies which generally export the production to other places.

**Table 5.** Opinions on the channel which can most improve the economic viability of farms.

| Supply Chains | Málaga | Granada |
|---|---|---|
| **Potential short chains (%)** | | |
| On-farm sales | 14.3 | 13.3 |
| Box schemes | 11.9 | 33.3 |
| Internet sales | 4.8 | 0 |
| Direct sales: restaurants and shops | 2.4 | 6.7 |
| Farmers' markets | 33.3 | 13.3 |
| **Potential long chains (%)** | | |
| Sales to intermediaries | 33 | 20 |
| Wholesale market (*Mercas*) | 14.3 | 13.4 |

Source: created by the authors.

### 4.3. Main Barriers to Scaling up SFSCs

In order to identify the factors which currently hamper the scaling up of the SFSCs, the following figures represent the evaluations made by the producers interviewed concerning the inconveniences which exist for the commercialization of their produce using SFSCs. For Malaga (Figure 1), the most relevant factors are the need to differentiate themselves from the big organic farms (56.1%) and the

distribution costs associated with the SFSCs (49.2 %), while in Granada (Figure 2), the main obstacle which has been identified is the lack of marketing strategies used to get to know their produce (33%). If we consider the grouping of all the sample data on the problems indicated as being highly relevant, it is important to highlight the fact that they need to differentiate themselves from other organic companies and producers (38.05%) despite the fact that this deals with a smaller group of agents. Next in importance is the low level of interest shown by local consumers and insufficient demand (29.6%), distribution costs (28.1%), regulatory barriers (27.65%) and the lack of marketing strategies being used (26.15%) Problems associated with logistics have an average score of 16.45%. In the survey, the producers were asked whether they would be willing to use regional logistics distribution platforms and more than half of those surveyed responded affirmatively (90%).

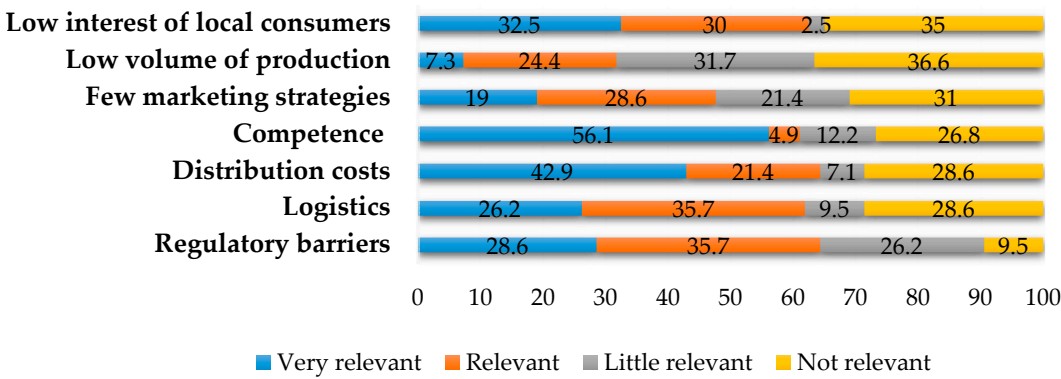

**Figure 1.** Factors that negatively affect the scaling up of short food supply chains (SFSCs) in Malaga. Source: created by the authors.

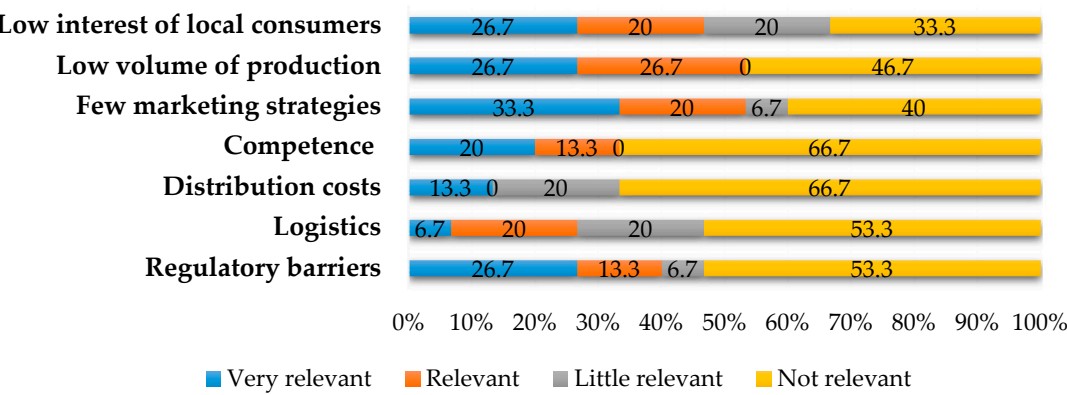

**Figure 2.** Factors that negatively affect the scaling up of SFSCs in Granada. Source: created by the authors.

Regarding services used to improve the business model for the supply chain, the global data from the producers' sample indicates, as has been previously shown, limited use of the internet and electronic commerce in general (Table 6). This is an obstacle to product visibility, to increasing the number of consumers and even improving efficiency in distribution, especially when the number of online purchases is constantly increasing. The use of brands and certification schemes can also be another way to add value by communicating the uniqueness and identity of their produce. Therefore, most of the producers interviewed in Malaga (96%) and Granada (76.9%) use eco-labels to identify their form of production better.

The lower percentage in Granada is related to the fact that there are some agro-organic producers who neither need nor want an identifying label, which they justify with the assertion that their production models are ones based on trust between consumers and producers. Another successful feature which can be identified, is that more than half of the producers in the sample work together to

commercialize their produce, which indicates the great importance of collective structures and social innovation after years of integration and collaboration between local agents. In Malaga, it is even more successful; the Guadalhorce organic cooperative groups together small-scale producers from the Guadalhorce Valley, on farms between 2 and 4 hectares, who cultivate organic fruit and vegetables.

**Table 6.** Services and activities used for local sales.

|  | **Málaga** | **Granada** |
|---|---|---|
| **Services used (%)** | | |
| Own website | 11.9 | 40 |
| Internet deliveries | 0 | 13.3 |
| Social media | 7.1 | 40 |
| Outsourced service | 9.5 | 6.7 |
| **Branding and Labelling used (%)** | | |
| Eco-labels | 96 | 76.9 |
| **Networks used for sales %** | | |
| Cooperatives and platforms | 52.4 | 66.7 |

Source: created by the authors.

In order to improve the knowledge of the agents of the food chain at a local scale, a series of closed questions were asked to small shops. As has already been indicated, this data is of interest, as both sectors (producers and retailers) are strategic in ensuring the sustainability of the regional agri-food system (Figures 3 and 4).

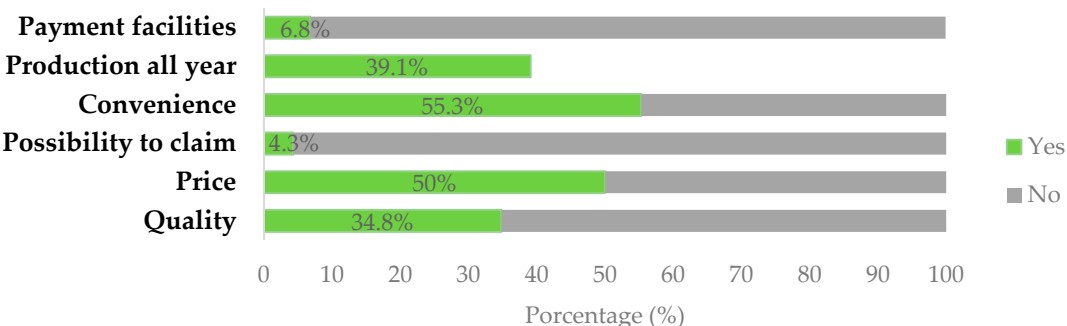

**Figure 3.** Purchasing at wholesale markets: aspects which are most highly valued by retailers in Malaga. Source: created by the authors.

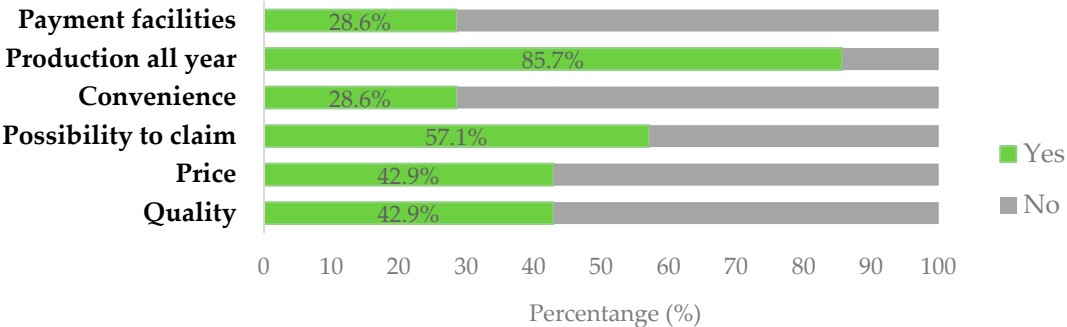

**Figure 4.** Purchasing at wholesale markets: aspects which are most highly valued by retailers in Granada. Source: created by the authors.

The majority of those interviewed in the subsample of small shops positively understand and value local produce because of its quality and freshness (93.8% in Granada and 95.7% in Malaga) and more than 80% would be willing to participate in an advertising campaign for local organic produce. When the small shops are asked to choose the way in which they would prefer to be supplied with

produce from the farmers, in both Granada and Malaga, the preferred formula is for products to be taken directly to the establishment, reaching 60% in the small shops in Malaga and up to 87.5% in Granada.

Nevertheless, there is a growing tendency in Spain for small shops to supply themselves directly in the wholesale markets (*Mercas*), instead of direct supply which the small producers have traditionally provided. In Malaga, small shops are mainly supplied through wholesale markets (83.5%), while in Granada, it is a little more than half (50.1%) Among the reasons given to justify the purchase in wholesale markets, the most common is the convenience of purchasing everything at one sales point (55.35%), followed by being able to choose the price of the produce being acquired (50%) (Figure 3).

The main reason given by small shops in Granada for preferring to purchase produce in wholesale markets is the year-round guarantee of availability of the quantity needed and product diversity (85.7%) and, secondly, that they can make a claim if there are any problems (57.1%).

## 5. Discussion

Metropolitan areas show enormous potential for organic producers involved in food chains operating in peri-urban areas. Investing in smarter logistics can shorten the distance between producers and consumers, stimulating market opportunities for local farmers and giving citizens access to fresh, healthy, and sustainably grown food [55]. Nevertheless, we have found in this study the difficulty in obtaining up-to-date statistical information that allows knowing the actual situation and evolution of small-scale organic farms that distribute mainly through these channels.

Our research highlights that the majority of the horticultural producers who sell using the SFSCs use different types of sales channels such as farm sales, box schemes, internet sales, direct sales restaurants and shops, farmers' markets and public procurement. It is important to highlight that diversification in commercialization methods contributes to reducing risk and to better meeting the demands and needs (time and place) of establishments and urban consumers. Nevertheless, despite the wide variety of channels used, the volume of sales continues to be very low, and the time and energy costs are high as a result of the dispersion of the sales points. This must be because, in general, the small producers have insufficient access to distribution infrastructures for the produce processing and preparation for the end consumer. However, this is also because there is a lack of qualified labor available in this field. Another reason is the lack of insufficient public aid for encouraging the creation, reorganization and strengthening of the SFSCs.

After carrying out the literature review and the analysis of the results, we conclude that cooperation and the network of relationships between producers and between multiple stakeholders involved in the bioregional food chain need to be strengthened for the SFSCs to reach their full potential, moving forward in innovation. However, insufficient commercial organization with small shops, restaurants and public food procurement weakens the position of peri-urban agriculture in the regional agri-food system. This also reduces the consumption of produce which has been grown locally and sustainably. Moreover, food policies in most cities and metropolitan regions continue to be scarce and are often nothing more than anecdotal initiatives. Some actions, including public procurement policies, can be powerful drivers in advancing towards sustainable food systems that are currently underused in many regions. Direct sales to small shops (neighborhood supermarkets, specialized shops and greengrocers), to the hospitality sector and to public food procurement can improve the volume of sales, contribute to establishing a stable food chain, increase visibility and trust in organic produce and, finally, improve income. These are all aspects that other direct channels, such as farmers' markets and consumer groups, do not always achieve, as they have a more limited reach regarding product demand and the number of consumers that they can attract.

Thanks to the data obtained in this research, we can highlight some issues which we consider to be of interest for the improvement of the SFSCs. The main causes identified which affect the performance of the SFSCs in the two regions studied in order of importance were: (i) competition within the organic sector itself; (ii) the low level of interest in local organic produce shown by local

consumers; (iii) distribution costs, and (iv) lack of marketing strategies used by the farms themselves. These areas, in agreement with the bibliographical review which has been carried out (Section 2), must be addressed cooperatively in order to find innovative comprehensive solutions to optimize the SFSCs. This is essential because the investment and management capabilities of the small producers are limited, which affects their chances of becoming more efficient, and secondly, because cooperation allows small producers to respond to demand. This would mean they could provide a stable and varied source of year-round produce to big customers, such as those in the hospitality sector and collective catering. However, this requires greater and better organization between various agents to achieve more efficient management in the production itself and, especially, a certain level of innovation and investment in distribution and logistics to better adapt to local needs.

The research suggests that the benefits provided by the collaborative SFSCs are greater than those from individual actions [56]; some of these collaborative benefits are: (i) improved product range available to consumers; (ii) resource sharing amongst producers and processors; (iii) maintaining local food chain infrastructure (such as abattoirs); (iv) increased negotiating power for small producers; (v) reduced competition between small producers; and (vi) mutual support to combat isolation and stress. The cooperative focus can meet some of the needs indicated in the interviews with the small shops, which are, among others, improving the ability to guarantee the volume and diversity of year-round produce, optimizing processes so they can be supplied by just one provider, having an effective customer service structure and having a competitive price in the market. These needs will undoubtedly be met when material and human resources are optimized.

As we have shown in recent research, [21,22], metropolitan food clusters can be a good opportunity to co-design production systems according to the demands of the local market, carry out long-term joint planning, use shared communication strategies, improve the consumer commitment and open new sales channels, without having to become large-scale producers. The results from Granada and Malaga show that the food hubs are seen as an opportunity to support the skills which some small producers lack, such as improving knowledge on health regulations, business planning, pricing and accountancy, as well as increasing marketing strategies to increase the value of traditional products. However, at the same time, they can boost collaboration with restaurants and public authorities or secure the finance needed for equipment for value-added product innovation. Therefore, these models, which are supported by a management team that can help small producers increase their impact at a local/regional scale by capturing higher added value, produced through product anchoring strategies and transformation. The food hubs are a good alternative to the growing concentration of a limited number of retail companies.

Another of the major obstacles identified in the two regions studied is the insufficient demand for the weak consumer commitment to local organic produce. Therefore, several studies have looked into consumer preferences and purchasing behaviors regarding organic and local consumption. For example, research carried out by Watss et al. [56] on consumer engagement with alternative food networks showed that respondents wanted to support the local economy by shopping with small local retailers and buying local food but, at the same time, they value supermarkets for their convenience and wider range of products and prices. The study suggested that their tendency to favor conventional food networks seemed to be based on reliance, whereas their use of SFSCs tended to be based on trust. [56]. Other studies show that a large number of consumers are not willing to pay extra for organic produce [57]. Therefore, this is one of the major challenges which must be overcome by increasing education and awareness of local and organic produce and food systems with support from public administrational bodies. Marketing strategies must be improved by the sector itself through increasing the visibility of local, organic attributes. Most of the small producers interviewed in the two regions studied make limited use of social media and internet sales, either because they do not have the required communication skills or due to a lack of time and resources, and this issue has to be tackled.

E-commerce can complement other sales channels and can improve local produce visibility, which is going to be essential after the changes caused by the recent COVID-19 pandemic. In this

sense, collaborative SFSCs, such as food hubs, can support producers by promoting technological innovation and supporting the processes and the products they offer to make it more competitive in the local market, while satisfying the demands of a growing number of environmentally aware consumers. The organic management developed by small producers and their commitment to social, environmental and economic values at the local scale can be used as a differentiation strategy in opposition to industrial agriculture. Similarly, we can find the creation of collective regionalized brands that show the benefits generated by local organic agriculture in the conservation of biodiversity and the supply of ecosystemic services in general, as well as providing healthy food. An advantage of developing a collective brand is that products can retain their identity when sold through a variety of channels, such as e-commerce and retail outlets [58].

At the same time, the creation of governance structures and regionalized networks which are dependent on endogenous resources and the specific characteristics of the regions is an extremely important driver for promoting collective learning processes and information exchanges. There is a need for a new pact between policies and organized civil society (farmers' cooperatives, food activists, community-based organizations, NGOs and researchers) involved in the planning and innovation around the SFSCs.

Special attention is given to the implementation of cooperative business models as a way in which collective efficiency can be increased and social and economic innovation can be encouraged [34]. Increasing the ability of small producers to generate knowledge to adapt to the new demands of the urban market is an essential factor for improving their viability and favoring local development. This should not only be in terms of economic growth, but also from a perspective focused on constructing a new niche market by selling high-quality organic products and re-valuing local production with a low environmental impact. Another option for strengthening the visibility of the small organic producers who grow produce in the metropolitan belts is the use of internet-based solutions (apps, websites, social media, online shops, etc.), which allow new market opportunities to be stimulated.

Nonetheless, innovation around production differentiation strategies and technical solutions to improve SFSCs and local productions is not enough, all of this needs to occur alongside the development and design of appropriate systems of regulation and information, educational campaigns, and public support [59]. In order to meet these objectives, it is necessary to widen research focused on knowledge on the most suitable policies to cope with the challenge to secure urban food provisioning, for example, by removing barriers for public procurement of local and organic foodstuffs or by orienting research on how to reduce vulnerability to future supply problems triggered by new pandemics. From a systemic perspective, heritage-based solutions linked to landscape patrimony to increase the added value of local food production is another interesting line of research.

## 6. Conclusions

A paradigm shift from industrial agriculture to diversified agroecological systems is more urgent than ever [8]. As identified in the research, there are three key domains involved in building agri-food system resilience, which is more embedded in metropolitan contexts and capable of ensuring access to healthy, local fresh food. These include supporting and improving the scalability of collaborative SFSCs, ensuring the social and economic viability of organic peri-urban agriculture and, finally, facilitating access to land and food producing resources in urban bioregions. Furthermore, different strategies should be put in place to gradually shift away from trade-oriented agricultural policies that disadvantage small-scale producers or favor unsustainable agricultural practices [8]. Scaling up the volume of sales means that small organic producers must adapt to new forms of consumption, which is shown by the growing trend to source food using the internet, for which they must incorporate digital platforms as an essential element of their operation [60].

A range of actions have been identified to improve the optimization of cooperative farm business. Some of them include cooperative business innovation processes to increase the efficiency in the distribution, logistics, and use of resources and services to meet food safety regulations, while at the



same time favoring collective knowledge and increasing communication strategies to value organic and local production as suitable actions. All these recommendations have shown limited efficiency, unless they are addressed as part of a set of cooperative strategies at various levels among the different agents of the food chain. All these aspects are more urgent than ever and, moreover, when food safety is being affected (accelerated by the COVID-19 crisis). Consumers increasingly understand the need for food which generates local revenue that is healthier, which reduces its impact and guarantees territorial supply in times of increasing uncertainty. However, accepting collective responsibility is paramount, as it is unlikely that any single actor can achieve even modest steps towards sustainability, while local policy action has the power to provide potential seeds of transformative change [59].

In order to face the great challenge which the cities and the metropolitan regions face in the improvement of food security [61], together with associated issues such as urban sprawl, demographic concentration in urban areas, loss of biodiversity, climate change and reduction in traditional small-scale farms and their identifying features requires the application of a system-based approach. This requires engaging urban planning to ensure spaces for peri-urban arable lands along with food planning policies. It also means developing ad-hoc policies for the activation of agricultural spaces and to guarantee the running of food supply (using structural and commercial approaches), the conservation and activation of the ecosystemic services, and the increase in the quality and quantity of these in both the sector itself and its companies [62].

Increasing knowledge on the mechanisms which allow new farmers to improve their professional training and strengthen their business and social media skills is indispensable [63]. As stated by Matarán [64], the challenge of implementing major governance in agri-food planning policies must not be forgotten either. Nor, according to Mata [65], should we forget the challenge of implementing agri-food planning policies that promote the SFSCs and the fostering of agricultural landscapes at a local/regional scale with the objective of achieving increasingly embedded food systems. Finally, active involvement of policy makers is required in concert with civil society to consolidate the reconnection between the city and the countryside, between food and territories, ties that are the foundation for the future preservation of peri-urban agriculture and its cultural landscapes.

**Author Contributions:** Theoretically conceptualized and designed this study, C.Y.O.; data analysis, C.Y.O. and A.M.R.; data curation, Á.M.F.; writing—original draft preparation, C.Y.O.; writing—review and editing, C.Y.O., A.M.R., R.M.O. and Á.M.F.; methodology, C.Y.O. and A.T.R. All authors have read and agreed to the published version of this manuscript.

**Funding:** Part of this research has received funding from the Spanish project SAMUTER and from the European Agricultural Fund for Rural Development (EAFRD) and the Spanish Ministry of Agriculture, Fisheries, Foodstuffs and the Environment, in the call 2018, submeasure 16.1 within the framework of National Rural Development Programme 2014–2020.

**Acknowledgments:** The authors would like to thank Heliconia s.coop.mad, Fundación Agroterritori, Guadalhorce Ecológico, Guadalhorce Grupo de Desarrollo Rural, Union de Uniones de Agricultores y Ganaderos and the Agroecological Network of Granada (RAG) for the support provided for the questionnaires and for helping to prepare the analysis.

**Conflicts of Interest:** The authors declare no conflict of interest. The funders had no role in the design of the study; in the collection, analyses, or interpretation of data; in the writing of the manuscript, or in the decision to publish the results.

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
