# Peer review of "Peri-Urban Organic Agriculture and Short Food Supply Chains as Drivers for Strengthening City/Region Food Systems—Two Case Studies in Andalucía, Spain"

_land, doi:10.3390/land9060177_

Round 1

Reviewer 1 Report

Excellent paper. Also topical and timely. 

Please shorten sentences! No because they are too wordy, or too technical, but because the text is concentrated when it comes to information; desirable to spread it out with much shorter sentences.

I note that you do not address urban sprawl. Not necessary for this paper, but would be good to not that clearly this sector needs to engage with urban planning to ensure spaces for in-city and peri-city arable lands are retained. It is much harder to reclaim built-up land than to prevent the being built up in the first instance, through proper planning and zoning with food production in mind.

Author Response

Thank you very much for the review.

English has been reviewed.

We have shortened the following sentences.
Lines:75-77; 93-94; 75-77; 174; 180; 200, 230, 552; 587;629; 634;696

Reviewer 2 Report

Comments and suggestions:

  1. I find the word case in the topic as well as in methodology section a bit confusing. I connect the word automatically to case research method aiming to an in-depth, detailed examination of a single or very limited number of individual organizations. I suggest that you consider different wording and clarify the methodology section.
  2. The literature review section 2.1 is could be strengthened. I would like to see more clearly where the topics 2.1.1 - 2.1.4 are derived.
  3. I would also like to suggest the writer(s) to get better acquainted to concepts of supply chain management  (see eg. Chopra & Meindl (2012). Supply Chain Management)

Author Response

Thank you very much for your review:

We have done the following changes following your remarks:

  • We have considered using different wording for the word case in the topic as well as in the methodology section:
  • Line 312: we remove "case study" and we added "this region"
  • Line 449: we remove the word "In this case" we changed it for "In Granada"
  • Line 539: we remove the word "cases" and we added "Granada and Malaga"
  • Line 600: we change "case studies" for "two regions studied".

We clarify the methodology section:

  • We removed the sentence 356 that was confusing
  • In line 370 we clarified where we gathered the data
  • In line 379 we clarified a sentence where we provided statistical information
  • We added en table 1 and table 2 more information related to the sample
  • In line 417 we clarified the information related with the questionnaire.

We strengthened the literature review section 2.1  by adding new references and clarifying the information that we had concluded in other papers related to this one:

Todorovic, V., Maslaric, M., Bojic, S., Jokic, M., Mircetic, D., & Nikolicic, S. Solutions for more sustainable distribution in the short food supply chains. Sustainability, 2018, 10(10), 3481.

Méndez, R. Renovar economías urbanas en crisis: un debate actual sobre la innovación. Desenvolvimiento Regional em debate, 2016, 6 (3), 4-31.

Swagemakers, P.; Domínguez, M.D.; Milone, P.; Ventura, F. and Wiskerke, J. Exploring Cooperative Place-Based Approaches to Restorative Agriculture. Journal of Rural Studies 2019, 68, 191–199. doi:10.1016/j.jrurstud.2018.12.003.

Mata-Olmo, R. Agricultura Periurbana y Estrategias Agroalimentarias en las Ciudades y Áreas Metropolitanas Españolas. Viejos Problemas, Nuevos Proyectos. In Cultura Territorial e Innovación Social. Edited by N. Barón and J. Romero. Valencia, España: Estudios y documentos, 2018, pp. 369–389.

Yacamán, C.; Mataran, A.; Mata, R.; López, J.M and Fuentes-Guerra, F. The Potential Role of Short Food Supply Chains in Strengthening Periurban Agriculture in Spain: The Cases of Madrid and Barcelona. Sustainability 2019, 11 (7): 1–19. doi:10.3390/su11072080

Yacamán, C.; Mata, R., and Matarán, A. Los Sistemas Territoriales de Innovación Social para Mejorar la Viabilidad de la Agricultura Periurbana. El Caso de Estudio de La Región Urbana de Madrid. In Desafíos y Oportunidades de un Mundo en Transición; Una Interpretación desde la Geografía. Edited by Farinós, J. (coord.), Valencia, España: Universidat de Valencia, 2020, 317-330.

Reviewer 3 Report

I found this to be a very interesting and well-presented paper based on sound methodology and on a topic of greater significance with the COVID-19 crisis.

I can highly recommend publication but with just a few minor amendments.

  • The COVID-19 crisis has not impacted international food supply chains as some feared. For example, here in the UK fresh produce from Spain has been arriving OK whereas UK sourced flour was in short supply as factories were not geared up to produce a higher volume of small bags of flour.
  • However, the COVID-19 crisis might lead to nations like the UK highly dependent on food imports to reduce reliance on Spain thus maybe inducing less demand for industrial food production in Andalucía.
  • In the UK the SFSC sector has been very badly hit by restaurant closures, notably dairying and cheese, but in the study area the restaurant trade does not seem to be so important. Is this because of the type of produce being less distinguishable than say artisan cheese?
  • In my/our studies of SFSCs in France and to some extent in the UK, wine and in the UK beers ciders and spirits were key products. But this does not seem to be the case in the study areas, though I know that apart from Malaga wine this part of Spain is not a major wine producer.
  • I prefer the reference system that uses (Smith (2020) in a study of…used….to demonstrate that……).This system has two advantages it places the authors in the readers eyes and tells us what the authors did. My request is that the references are changed but the current system might be the journal’s house style.
  • A few English lapses but overall excellent English. A couple of examples: In line 399 increases should be increase and in line 663 this should be these.
  • Figures 3 and 4 need Malaga or Granada in the title.
  • There is not much discussion as to whether low consumer interest is due to a price difference and related to convenience/time. In other words, are consumers put off by an assumption (not always true) by consumers that organic and SFSC food is more expensive and/or because busy consumers want to do one big shop per week in one place. Certainly, the Exeter Farmers Market in South West England is dominated by wealthy retirees.
  • For the future maybe a study of consumers of SFSCs by customer type. For example, do wealthy retired residents from northern Europe with maybe more time and inclination to buy organic local food form a disproportionate proportion of consumers?
  • Likewise, do niche hotels and restaurants provide a greater ratio of consumers, Do the famous Paradors make a point of using SFSCs in their publicity, assuming that they do use organic SFSC foods.
  • The references are comprehensive but might be helped by including some of the work done by British authors, for example Ilbery, Gilg, Marsden, Kneafsey, Maye et al. and some of the volumes in the Perspectives on Rural Policy and Planning.
  • Finally, locked out of Europe (here in the UK) the paper was a joy to read as it reminded me of several University field trips to the Costa del Sol and excursions to Granada and the major dam which made so much of the horticulture possible.

Author Response

Thank you very much for your observations.

  • We change some of the reference style as suggested in line:

    62-63, 111, 152, 160, 182, 187, 199, 211, 224, 233, 279, 285, 298, 729, 730

  •  English lapses changed in line 399  and in line 663.
  • We added Malaga and Granada in the title of figures 3 and 4
  • We added more discussion on consumer interest and we added new references as suggested:

    Watts, D., Little, J., and Ilbery, B. ‘I am pleased to shop somewhere that is fighting the supermarkets a little bit’. A cultural political economy of alternative food networks. Geoforum 2018, 91: 21–29. doi.org/10.1016/j.geoforum.2018.02.013

    Denver, S., Jensen. J. D., Olsen, S. B., and Christensen, T. Consumer Preferences for ‘Localness’ and Organic Food Production. Journal of Food Products Marketing 2019, 25 (6): 668–689 doi.org/10.1080/10454446.2019.1640159.